# The Ultimate (Mis)match: When DNA Meets RNA

**DOI:** 10.3390/cells10061433

**Published:** 2021-06-08

**Authors:** Benoit Palancade, Rodney Rothstein

**Affiliations:** 1Institut Jacques Monod, Université de Paris, CNRS, F-75006 Paris, France; 2Department of Genetics & Development, Columbia University Irving Medical Center, New York, NY 10032, USA

**Keywords:** DNA repair, genetic recombination, genetic stability, transcription, RNA, ribonucleotide, DNA:RNA hybrid, R-loop

## Abstract

RNA-containing structures, including ribonucleotide insertions, DNA:RNA hybrids and R-loops, have recently emerged as critical players in the maintenance of genome integrity. Strikingly, different enzymatic activities classically involved in genome maintenance contribute to their generation, their processing into genotoxic or repair intermediates, or their removal. Here we review how this substrate promiscuity can account for the detrimental and beneficial impacts of RNA insertions during genome metabolism. We summarize how in vivo and in vitro experiments support the contribution of DNA polymerases and homologous recombination proteins in the formation of RNA-containing structures, and we discuss the role of DNA repair enzymes in their removal. The diversity of pathways that are thus affected by RNA insertions likely reflects the ancestral function of RNA molecules in genome maintenance and transmission.

## 1. Introduction

Among the many scientific contributions that Miro Radman has made to our understanding of genome biology, his work on the SOS response and mismatch repair stands out amongst the most visionary. Combining observations that he and his colleagues made using bacteria and their phages, he outlined the main features of the cellular response to DNA damage and foreshadowed its importance for genetic stability [1,2,3,4]. An important concept arising from these pioneering studies is that damage to the genome can be tolerated by calling into play the cellular machineries that ensure viability, even at the expense of genetic integrity.

During the past several years, an increasing number of obstacles to DNA-related transactions was found to similarly trigger DNA repair and tolerance mechanisms. Among them, RNA-containing structures have recently caught the attention of DNA biologists, as addressed in several recent excellent reviews [5,6,7,8,9,10]. These range from single ribonucleotide insertions to RNA stretches, DNA:RNA hybrids and R-loops, in which a single-stranded DNA is displaced (Figure 1). Such structures are observed in diverse species and represent a significant fraction of their genomes: for example, ribonucleotides are incorporated in newly synthesized DNA at an overall rate of ~1:1000 nucleotides [11] and R-loops occupy approximately 5% of the human genome [12]. It has become increasingly clear that these structures are relevant for genome integrity: on the one hand, genetic screens have highlighted the importance of RNA metabolism factors in the DNA damage response [13,14,15,16], and on the other hand, factors classically associated with DNA repair or genome maintenance handle RNA-containing substrates in vitro [17,18,19,20]. Finally, dedicated enzymes can specifically process DNA/RNA hybrid structures, as exemplified by the H class of nucleases (RNase H1 and H2 in eukaryotes), which hydrolyze RNA moieties in DNA:RNA duplexes [21].

Here, we review evidence obtained from in vivo studies in light of findings arising from in vitro experiments and summarize: (i) how cellular enzymes involved in DNA metabolism recognize, utilize or tolerate RNA-containing substrates, and (ii) how this substrate promiscuity can either trigger the generation of genotoxic structures, or impact their repair.

## 2. When DNA Polymerases Meet Ribonucleotides

### 2.1. DNA Polymerases Can Use rNTPs during DNA Synthesis

Since DNA polymerases are generally unable to catalyze *de novo* DNA synthesis, the initiation of DNA replication requires the activity of DNA-dependent RNA polymerases, or primases, which utilize ribonucleotides triphosphate (rNTPs) to synthesize short, ~10 nt-long RNA primers, providing a free 3′ hydroxyl for further leading and lagging strand elongation. Notably, replicative DNA polymerases (Pol α, Pol δ, and Pol ε), despite having a steric-gate residue favoring dNTP selection at their nucleotide binding pocket, can also incorporate rNMPs into newly synthesized DNA, as highlighted by in vitro studies ([11]; Figure 1a). Even with rNMP incorporation disfavored, the high cellular concentration of rNTPs compared to dNTPs results in rates of ribonucleotide misincorporation within nascent DNA from ~1:600 to ~1:5000 depending on the DNA polymerase considered [11,22]. Thus, rNMP insertion represents a major error that occurs during DNA replication, which has been confirmed by genome-wide mapping [23].

### 2.2. Recognition of Embedded Ribonucleotides: Genotoxicity vs. Tolerance

The insertion of rNMPs in the genome enhances DNA reactivity, favors alkali cleavage, and causes backbone distortions, with possible repercussions on DNA-related transactions, such as replication and chromatin assembly [24]. In view of their potentially deleterious effects, genome-embedded rNMPs are efficiently removed by ribonucleotide excision repair (RER), an error-free pathway that relies on ribonucleotide recognition and excision by RNase H2, which also removes the RNA stretches generated by primase activity [24]. Alternatively, in the absence of RNase H2, ribonucleotide insertion leads to topoisomerase I-dependent cleavage, eventually leading to short deletions in repeated sequences, as revealed by in vitro and in vivo studies [25,26]. Furthermore, the accumulation of rNMPs insertions in the genome causes replication stress and genetic instability, as shown by the phenotypic analysis of RNase H2 loss-of-function or pol ε steric-gate mutants [25,27,28]. Post-replication repair pathways are also required to tolerate rNMPs insertions within the genome. Notably, translesion synthesis by Pol ζ can utilize rNMP-containing templates, as confirmed by in vitro studies [27].

### 2.3. How Ribonucleotides Insertions Benefit Genome Homeostasis

Although the insertion of ribonucleotides and their processing by the DNA metabolism machinery can be detrimental for genetic stability, they can benefit genome homeostasis in some situations (Figure 1a). For example, some non-replicative polymerases, such as Pol μ, exhibit atypically high rNMPs incorporation rates [29]. Importantly, Pol μ activity is required to fill DNA overhangs before ligation in the non-homologous end joining (NHEJ) pathway. Thus, ribonucleotide incorporation by Pol μ favors the action of DNA ligase 4, which tolerates mispairs or gaps in the opposite strand, yet in this case prefers a terminal ribonucleotide [29]. Another DNA repair pathway that reportedly uses embedded ribonucleotides is mismatch repair (MMR), which is particularly critical to correct nucleotide misincorporations that occur during replication. Unrepaired rNMPs, whose insertions are naturally restricted to newly synthesized DNA, likely mark the DNA strand in which mismatches have to be corrected. Indeed, RNase H2 is required in vitro and in vivo for mismatch repair in the leading strand [30,31], supporting a model in which recognition and cleavage at embedded ribonucleotides provides a strand-specific entry site for the MMR machinery. Whether other DNA repair pathways, beyond NHEJ and MMR, similarly involve one or multiple enzymatic activities with a shared preference for rNMPs remains to be determined. Of note, the presence of a few embedded ribonucleotides enhances the in vitro resection activity of EXO1, a nuclease involved in the processing of DNA double-strand breaks (DSB) [32], supporting the existence of additional cellular activities preferably handling rNMP-containing substrates.

## 3. R-Loop Formation: Is the Homologous Recombination Machinery Invited?

The presence of RNA moieties within genomes is not limited to embedded ribonucleotides or RNA primers introduced by the replication machinery. In addition, RNA molecules transcribed by RNA polymerases can anneal to their complementary DNA following their synthesis, generating DNA:RNA hybrids with displaced ssDNA strands, or R-loops (Figure 1b). This process is favored by the negative super-coiling of the DNA helix behind the transcribing polymerase and the G-richness of the RNA moiety, but counteracted by RNA folding and ribonucleoparticle assembly [6]. While unscheduled accumulation of R-loops leads to genomic instability, their programmed formation can contribute to epigenetic or transcriptional regulation [6,7]. Strikingly, although R-loops can form spontaneously upon in vitro transcription of R-loop-prone sequences [33], a growing body of evidence supports the notion that their generation involves dedicated enzymatic activities.

### 3.1. R-Loop Formation: Are There Enzymatic Activities Involved In Vivo?

R-loops are mostly believed to form co-transcriptionally, whereby the RNA molecule synthesized by an RNA polymerase hybridizes to its DNA template in *cis*. Furthermore, specific reporter systems have been designed to address whether R-loops can also form in *trans.* In that case, the RNA released from its transcription site would anneal to an homologous sequence elsewhere in the genome. Koshland and colleagues designed an *S. cerevisiae* strain, in which the same sequence is carried by both chromosome III and a yeast artificial chromosome (YAC). DNA:RNA hybrid immunoprecipitation (DRIP) using the S9.6 monoclonal antibody, which binds hybrids with high affinity, revealed that transcriptional induction from the chromosomal locus triggered DNA:RNA hybrid formation on the YAC homologous sequence, suggesting that the corresponding RNAs can form R-loops in *trans* [34]. More recently, Lingner and colleagues reported in human cultured cells that plasmid-borne expression of telomeric repeat-containing RNAs (also known as TERRA) leads to their recruitment at telomeres through DNA:RNA hybrid formation in *trans* [17]. Strikingly, in these situations, the formation of DNA:RNA hybrids requires the homologous recombination (HR) protein Rad51 [17,34] and furthermore, in human cells, depends on its strand-exchange activity ([17]; Figure 1b).

Does the formation of endogenous R-loops at genomic loci also require Rad51? Naturally-occurring telomeric R-loops were found to similarly depend on Rad51 in human cells, as shown by DRIP [17]. In both budding and fission yeasts, the spontaneous accumulation of R-loops detected in certain RNA metabolism mutants was suppressed in the absence of Rad51 as well as Rad52, an important Rad51 mediator protein [34,35]. On the other hand, R-loops scored upon inactivation of the mRNA biogenesis pathway (*tho* mutants), or upon simultaneous inactivation of RNase H1 and RNase H2, do not require Rad51 activity for their formation [34,36]. Of note, these reports have used distinct R-loop detection assays, including immunofluorescence with the S9.6 antibody, an approach reportedly suffering from specificity issues [37]. Thus, complementary studies are required to assess whether Rad51 is generally required for R-loop formation at genomic loci in *cis*, and how it impacts the formation of R-loops at repeated sequences in *trans*.

### 3.2. Proposed Mechanisms for How HR Proteins Could Favor R-Loop Formation

Rad51 was found in complex with TERRA or certain mRNAs in vivo– [17,38], and in vitro binding assays support the idea that this association is direct and involves its ssDNA binding activity [17]. These observations raise the possibility that Rad51 catalyzes the invasion of the RNA into the duplex DNA in a mechanism resembling the reaction occurring during homology-directed repair (HDR). Consistent with this view, in vitro assays revealed that pre-incubation of human Rad51 with TERRA RNAs promotes invasion of a circular dsDNA template, which depends on the strand-invasion activity of Rad51 in the so-called forward strand-exchange reaction ([17]; Figure 2a). Since the same result was not observed in another similar assay [18], further in vitro experiments are necessary to identify the requirements for this Rad51 strand-exchange activity and reconcile these conflicting results. In this view, it is likely that the sequence of the RNA substrate and the topology of the dsDNA template influence R-loop formation in such assays. Furthermore, although Rad52 is unable to perform this reaction in vitro [39], the ssDNA-binding protein complex, RPA, also exhibits R-loop forming activity in vitro on a circular dsDNA template in the forward strand-exchange assay [18], suggesting that additional proteins of the HR machinery should be tested for their contribution to R-loop formation in vivo.

An alternative proposed mechanism for R-loop formation could involve an inverse strand-exchange reaction, in which binding of a recombinase to dsDNA favors strand exchange with a free complementary RNA template, an activity displayed in vitro by the bacterial recombinase RecA [40,41] and by Rad52 ([19]; Figure 2b). However, the fact that Rad51 proteins do not support the inverse strand-exchange reaction in vitro [19], while being required for R-loop formation in vivo, favors the forward model.

A third possibility may involve Rad51 DNA strand exchange activity during a classical HDR reaction, where the displaced ssDNA could anneal to complementary RNAs, as previously hypothesized [17,34]. In the future, identifying whether specific cofactors are important for RNA-dependent Rad51 activity will certainly pave the way to a better understanding of the catalytic reactions underlying R-loop formation. In this respect, it should be noted that these processes may also involve activities outside the HR machinery, as supported by the described DNA-RNA strand exchange activity of the polycomb repressive complex PRC2 in vitro [42], and by the reported requirement of RNA helicases (i.e., DHX9, DDX1) for unwinding RNA secondary structures prior to R-loop formation in vivo [43,44].

### 3.3. Addressing the Impact of HR-Dependent R-Loops on Genome Functions

Although there is increasing evidence supporting a role for the HR machinery in contributing to R-loop formation, it is still an open question as to how this process impacts genome integrity. Rad51-dependent genetic instability was reported for yeast artificial chromosomes or human telomeres forming R-loops in *trans* [17,34]. A possible interpretation of these observations is that Rad51 activity itself is responsible for the generation of R-loops in *trans*, resulting in the observed genetic instability. An alternative interpretation, which is not dependent on Rad51 R-loop forming activity, is that the R-loops themselves trigger DNA damage that is channeled into Rad51-dependent HDR events, leading to genetic instability. In agreement with the latter possibility, existing evidence supports the idea that R-loop formation can precede Rad51 binding. For example, transcription stimulates Rad51 recruitment to YAC sequences [34], and R-loop formation is a prerequisite for Rad51 binding to shortened telomeres, whose length maintenance involves the HDR pathway [45]. Understanding whether the HR machinery contributes to the generation of genotoxic R-loops in some situations will certainly require genetic tools to disentangle its function in hybrid formation from its role in hybrid-dependent DNA damage repair. Progress in this direction has been made utilizing a Rad51-independent single-strand annealing assay, which revealed that R-loop-associated genetic instability mainly arises from RNA produced in *cis* and does not require Rad51 activity [36]. Importantly, beyond their impact on genome integrity, R-loops formed in *trans* could also regulate transcription of their target loci, as reported in yeast and plants [46,47]. Whether the HR machinery contributes to R-loop formation in the context of coordinated transcriptional regulation remains to be determined.

## 4. R-Loop Resolution: Where DNA Repair Enzymes Come to Play

While the intrinsic ability of HR proteins to handle nucleic acid molecules can contribute to the formation of R-loops, a number of DNA repair factors have also been shown to recognize these structures, contributing to their removal or processing them into damage.

### 4.1. Direct and Indirect Roles of DNA Repair Factors in R-Loop Dissociation

The Fanconi Anemia (FA) pathway involves a number of factors that cooperate to remove replication-blocking lesions, such as inter-strand crosslinks, ensuring genome integrity during cell cycle progression. Strikingly, several FA proteins, i.e., BRCA1, BRCA2 and FANCD2, are recruited to transcribed, R-loop forming regions, and their inactivation triggers the accumulation of R-loops and R-loop-dependent damage [48,49,50,51]. While FA proteins may directly recognize R-loops, as shown in vitro for FANCI-FANCD2 [52], their activity at sites of transcription-replication conflicts may contribute to R-loop resolution through different mechanisms (Figure 2c–e).

The branchpoint translocase activity of FANCM, which was previously reported to target replication forks and Holliday junction intermediates, can dissociate DNA:RNA hybrids in vitro [51,53,54]. Although the canonical FA pathway is absent in budding yeast, inactivation of Mph1, a FANCM-related helicase, leads to R-loop accumulation and co-lethality with RNase H inactivation [55,56]. Together with the observation that Mph1 dissociates R-loops in vitro [53], these data support a conserved function in R-loop resolution for branchpoint translocases.

Other R-loop-removing activities are recruited through the FA pathway, as exemplified by BRCA2 association with RNase H2 [57]. In addition, BRCA1 and BRCA2 interact with SETX and DDX5, respectively, two DNA:RNA hybrid unwinding helicases, likely favoring their activity at the sites of hybrid formation ([50,58]; Figure 1b). Finally, members of the FA pathway were also reported to associate with BLM, a RECQ-helicase involved in the resolution of concatenated DNA molecules at replication forks. Strikingly, inactivation of BLM in human cells, or of its counterpart Sgs1 in yeast, triggers R-loop accumulation and R-loop-dependent genetic instability [59]. Of note, both BLM and Sgs1 unwind R-loop substrates in vitro, with the same efficiency as D-loops [59,60].

A number of FA-associated factors thereby participate in R-loop removal through different enzymatic mechanisms. How these distinct activities are coordinated and whether they also contribute to R-loop resolution independently of replication are still open questions.

### 4.2. R-Loop Processing by DNA Repair Enzymes

The R-loop structure can also be recognized by different DNA repair enzymes with single-strand endonuclease or DNA-modifying activities, leading to DNA incision or modification. In most cases, inactivation of such enzymes leads to R-loop accumulation, yet suppresses R-loop-dependent damage [61,62,63,64]. In this setting, the XPF and XPG endonucleases, which are well characterized for their role in excising single-strand DNA patches during the course of Nucleotide Excision Repair (NER), were the first enzymes proposed to process R-loops into genotoxic intermediates. XPF and XPG trigger genetic instability when R-loops accumulate following the inactivation of several R-loop preventing factors, including SETX and AQR helicases, or topoisomerase I, or upon transcriptional induction by estrogen [61,64,65]. It is likely that XPF and XPG recognize within the R-loops the same dsDNA-ssDNA junctions that they target in the NER pathway. Consistently, XPF and XPG introduce single-strand breaks within in vitro formed R-loops [66], although additional factors, including the Transcription-Coupled Repair (TCR) proteins XPA, XPB, CSA and CSB, contribute to their R-loop processing function in vivo [61].

A growing number of enzymes associated with distinct DNA repair pathways were similarly proposed to target ssDNA within R-loops, thus mediating their damage-triggering potential (Figure 1b). The Flap-endonuclease FEN1, which functions in Okazaki fragment processing and Base Excision Repair (BER), and to a lesser extent, the nuclease MRE11, which mediates double-strand break (DSB) resection, cooperate with XPF/XPG to generate DNA breaks at R-loops formed upon topoisomerase I inhibition in human cells [64]. In addition, the CtIP endonuclease, another DNA end processing factor, was also shown to prevent R-loop accumulation and to stimulate ssDNA break formation in mammals, a function possibly conserved with its yeast orthologue, Sae2, whose inactivation exhibits genetic interaction with R-loop-resolving factors [63]. Finally, the MutLγ complex, the endonuclease component of the MMR pathway, causes R-loop-dependent genetic instability at triplet nucleotide repeat (TNR) loci in budding yeast [62]. However, in most of these situations, the cleavage of R-loop substrates by the corresponding nucleases has not been reconstituted through in vitro assays, and the contribution of their enzymatic activities in R-loop processing was not evaluated in vivo. Such an assessment is all the more important since there is a case where a non-nucleolytic role has been assigned to the Mre11 nuclease in R-loop regulation [55].

Beyond nuclease-mediated cleavage, the ssDNA exposed by R-loops can also be targeted by base-editing enzymes. In yeast, expression of distinct mammalian cytidine deaminases, i.e., AID and APOBEC3B, triggers R-loop-dependent mutagenesis with a bias for the non-transcribed strand [67,68]. In addition, the native yeast cytosine deaminase, Fcy1, is recruited to R-loop-forming TNR loci and contributes to their fragility. Fcy1-mediated cytosine deamination to uracil channels these loci towards the BER pathway, where the sequential action of the uracil glycosylase Ung1 and the abasic site endonuclease Apn1 cause TNR contractions [62]. Of note, in vitro studies confirmed that an abasic site within a synthetic R-loop is targeted by APE1, the major human apurinic endonuclease [69].

### 4.3. DNA Repair Enzymes Encountering R-Loops: A Double-Edged Sword?

R-loop processing by the aforementioned nucleases may result in erroneous, unscheduled generation of DNA damage on R-loop forming sequences, accounting for R-loop-associated mutagenesis and genetic instability. However, ssDNA cleavage events may be part of natural R-loop clearance pathways ultimately favoring DNA repair and genome integrity. Taking this view, cleavage could release structural constraints associated with R-loop formation and further expose ssDNA ends, providing the entry point for R-loop-resolving enzymes acting downstream in R-loop removal. Supporting this hypothesis, telomere fragility caused by FEN1 inactivation is mitigated by hybrid degradation [70], and the sensitivity to genotoxic stress caused by Sae2 or CtIP loss-of-function is alleviated when DNA:RNA hybrids are unwound by Sen1/SETX [63]. Such observations suggest that these nuclease activities are critical to resolving R-loop-containing intermediates that would otherwise compromise genetic stability and survival. Similarly, repair of DSBs within transcriptionally-active regions can involve R-loop cleavage by XPG, which channels repair toward homologous recombination [71]. Finally, RNA-dependent genetic instability is utilized in host defense mechanisms for at least two well-characterized situations. In bacteria, endonucleases of the CRISPR-Cas system use short RNAs as guides to cleave exogenous DNA molecules [72]. In mammals, targeting AID to R-loops is part of the programmed genomic rearrangements occurring in class switch recombination during the immune response, although the lack of strand-specific effects suggest the existence of additional mutagenic mechanisms (reviewed in [73,74]).

While R-loop recognition and processing by DNA repair factors can be beneficial for genome homeostasis, it is unclear which parameters dictate the choice of the nuclease(s) or modification enzyme(s) that will preferentially engage in a given situation. In view of the preference of some enzymes (e.g., AID, [75]) for structured substrates, it is tempting to speculate that the sequence, the ssDNA secondary structure and the topological constraints associated with R-loops are features that define their processing.

## 5. DNA:RNA Hybrids in DNA Repair: Scaffolds or Obstacles?

### 5.1. DNA:RNA Hybrids Accumulation as a Consequence of DNA Damage

As summarized above, DNA damage can result from the accumulation of diverse RNA-containing structures, including DNA:RNA hybrids. Strikingly, the last decade has also revealed that DNA:RNA hybrids accumulate in *cis* at DSBs caused by reactive oxygen species, laser irradiation or site-specific endonucleases, as observed in several distant eukaryotes from yeast to metazoans [5,10]. In this context, the hybrids have been proposed to form as a consequence of de novo transcription initiating at DNA ends. Accordingly, RNA synthesis at DSBs is associated with increased recruitment of RNA polymerase II [76,77,78], with reported contributions from RNA polymerase III [79] or RNA polymerase IV in plants [80]. In this view, DNA ends bound by MRN (MRE11-RAD50-NBS1) function as promoters and assemble canonical pre-initiation complexes, resulting in bidirectional transcription from the DSB site [78]. Annealing of the newly synthesized RNA to the 3′ DNA overhang generated by resection then forms double-stranded DNA:RNA duplexes at the break (Figure 1c). Since hybrid accumulation has been preferentially detected at DSBs located in transcriptionally-active regions [81], it is also likely that changes in the dynamics of ongoing transcription contribute to the accumulation of canonical three-stranded R-loop structures engaging the pre-existing RNAs (Figure 1c). There is also evidence that in metazoans, DSBs trigger the formation of small RNA species arising from the processing of double-stranded RNAs, which themselves result from the hybridization of de novo and pre-existing transcripts near the break site [5]. Interestingly, a novel species of small single-stranded RNAs was recently reported to contribute to DNA repair in human cells [82]. In line with the accumulation of various types of RNA and DNA:RNA hybrid species, DSBs recruit a growing number of RNA and R-loop processing factors, such as the exosome, the splicing machinery, RNase H2 and Sen1/senataxin [57,81,83,84,85]. While the choreography of their respective recruitment will require further investigation, the network of RNA-dependent interactions formed at DSBs has the potential to influence their fate and repair.

### 5.2. Do DNA:RNA Hybrids Contribute to the Outcome of DNA Repair?

The physiological relevance of the RNA and DNA:RNA hybrid response at DSBs has remained controversial, possibly due to the different approaches used to address this question. Several groups have tackled the functional consequences of RNA synthesis and DNA:RNA hybrid accumulation at DSBs, by interfering, for instance, with transcription [78], RNA processing [86], RNA degradation [87], or hybrid accumulation [76,81,85]. In this setting, impairing de novo transcription or RNase H-dependent hybrid removal reportedly reduces DNA repair efficiency, as probed by HR or NHEJ reporters [76,79,86]. These studies support the idea that DNA:RNA hybrids are intermediates in DSB processing. In contrast, preventing hybrid unwinding through Sen1 or senataxin inactivation was reported to lead to increased NHEJ efficiency in yeast or human cells [81,85], suggesting that the presence of DNA:RNA hybrids modulates the DSB repair pathway choice. In the future, additional experiments using distinct reporter systems will be required to tackle the functional importance of DNA:RNA hybrid formation for DSB repair. In particular, it will be of paramount importance to provide novel tools to disentangle the multiple functions of the aforementioned factors, not only at the DSB sites but also elsewhere in the genome. Indeed, inhibition of transcription, or inactivation of the R-loop removing machinery, are expected to cause drastic changes in DNA and RNA metabolism, as exemplified by the transcriptomic changes that result from Sen1 or RNase H inactivation in yeasts [88,89].

### 5.3. Mechanisms by Which DNA:RNA Hybrids Impact the DNA Damage Response

How could the presence of DSB-associated DNA:RNA hybrids foster or impair DSB processing and repair? Hybrids were proposed to protect DNA 3′ overhangs [79], possibly preventing excess resection [76], while contributing to the recruitment of a number of DNA repair factors, including RPA, BRCA1, BRCA2, 53BP1 [57,58,76,78], with some of them possibly directly recognizing their structures. Consistently, in vitro reconstitution of *de novo* transcription events occurring at DNA ends revealed that local RNA synthesis is sufficient to promote the recruitment of DNA damage response factors, perhaps by forming foci that have liquid-phase separation properties [78]. In line with the reported role of RNA-protein interactions in granule formation, it is thus likely that the local accumulation of RNA species or DNA:RNA hybrids helps spatially organize DNA damage recognition, signaling and repair. Conversely, in some settings, hybrid accumulation has been shown to interfere with the recognition of DNA ends and to alter the dynamics of the resection process [85,87], in line with the observation that resection enzymes are differentially impaired by RNA-containing substrates in vitro [32]. Similarly, hybrid accumulation was shown to counteract the formation of Rad51 filaments [81], possibly explaining the increased NHEJ and decreased HR observed in this situation. Beyond these apparent discrepancies, it is tempting to speculate that hybrid formation is regulated according to the cell cycle stage or the level of genotoxic stress, to further modulate the efficiency and the accuracy of DSB repair.

## 6. When RNA Acts as a Template in DNA Repair

In addition to forming structures that impact DNA repair, RNA molecules can also serve as a source of genetic information to restore DNA sequences following damage. RNA-templated DNA synthesis had been observed for decades, but this activity was believed to be restricted to reverse transcriptases (RT) encoded by virions or retrotransposons, or telomerases. A growing body of evidence supports the view that RNA-dependent processes are also used in cells to modify undamaged DNA sequences, or to repair DSBs.

### 6.1. RNA-Templated Repair: Lessons from Yeast

Studies performed in budding yeast in the early 1990s provided the first evidence for the involvement of RNA intermediates in recombination. Garfinkel and colleagues introduced an artificial intron (AI) in the antisense orientation within a reporter gene, and further scored genetic modification events relying on transcription and splicing of the RNA intermediate [90]. Strikingly, the detection of recombinants strictly required the expression of endogenous Ty1 retrotransposons, suggesting that their intrinsic RT activity could somehow handle cellular RNAs and convert them into cDNAs used for recombination. Further studies suggested that Ty1 cDNAs can serve as the donor template for homologous recombination events targeting endogenous Ty1 sequences, or for non-homologous repair of inducible DSBs [91,92,93]. To specifically track the involvement of RNA intermediates in DSB repair, Storici and colleagues further modified the AI reporter system by introducing an inducible DSB within an homologous sequence [94]. By using variations of this reporter system, in which the template RNA is transcribed either at the inducible DSB locus (in *cis*), or from a remote sequence (in *trans*), they observed that the predominant repair pathway involves the synthesis of a cDNA intermediate. Indeed, most RNA-dependent repair events are suppressed in mutants impacting Ty1 expression, upon pharmacological or genetic inhibition of RT, and in *Saccharomyces paradoxus* strains lacking Ty1 elements [39,94].

Beyond cDNA-mediated recombination, RNAs can also be directly used as recombination templates, without reverse transcription. It was first reported that the introduction of RNA oligonucleotides within yeast cells allows for the repair of chromosomal, inducible DSBs, yet this process does not depend on cellular reverse transcriptase activities supplied by retrotransposons or telomerase [95]. In addition, the AI reporter system described above further scored RNA-dependent recombination events that do not require cellular transposition and increase after RNase H1 and H2 inactivation, supporting the existence of DNA:RNA hybrid intermediates [94]. Such RNA-templated repair occurs predominantly in *cis*, when the transcribed RNA is used to repair its own DNA sequence, resulting in precise repair of the DSB using the information from the RNA template [94].

### 6.2. Which Enzymatic Activities Are Necessary for RNA-Templated Repair?

The requirement for Rad51 and Rad52 homologous recombination proteins in cDNA-templated repair supports the notion that classical HDR mechanisms are involved ([39]; Figure 1d). In contrast, direct RNA-templated DNA repair only requires Rad52 as it is still observed in the absence of Rad51 or the Rad52 paralog, Rad59, or upon inactivation of resection nucleases or NHEJ factors [19,39,94]. How could Rad52 catalyze RNA-templated recombination? In vitro experiments revealed that Rad52 associates with a variety of RNA-containing substrates, including ssRNA, DNA:RNA hybrids, and synthetic R-loops [19,20,96]. One possible mechanism could thereby involve the formation of a Rad52-RNA complex that would further engage with a DNA duplex in a forward strand-exchange reaction, as observed in vitro ([20]; Figure 2a). Alternatively, a Rad52-dsDNA complex could interact with an RNA molecule in an inverse strand-exchange reaction (Figure 2b). Notably, both yeast and human Rad52 more efficiently promote inverse compared to forward strand-exchange in vitro [19,94]. Furthermore, neither Rad51 nor Rad59 displayed inverse strand-exchange activity in the same experimental setup [19]. Together with the genetic requirements of RNA-templated repair in vivo, these experiments support a model in which Rad52 inverse strand-exchange activity catalyzes the annealing of dsDNA ends to transcribed RNA.

Once annealed to DNA, RNA molecules could contribute to DSB repair either by bridging the DNA ends, as supported by in vitro experiments [20], or by providing a template that would be copied to elongate the DNA molecule. Since RNA-templated repair occurs in the absence of retrotransposon-encoded reverse transcriptase [39], the latter model is likely to involve cellular DNA polymerases. Although replicative DNA polymerases α and δ can handle RNA templates in vitro [95], RNA-templated-repair depends on the translesion polymerase Pol ζ in vivo ([39]; Figure 1d). Of note, in the system tested, Pol ζ is also required for RNA-templated DNA modification occurring even in the absence of induced damage (referred to as R-TDM in [39]). Whether this latter reaction involves spontaneous or catalyzed R-loop formation requires further investigation, notably since the intron harbored by the AI reporter may actually prevent hybrid formation [97].

### 6.3. RNA-Templated Rearrangements in Other Model Systems: Same but Different

While experiments performed in the budding yeast model have been instrumental in dissecting the genetic requirements of RNA-templated repair, mounting evidence suggests that related mechanisms exist in other organisms. Similar to the cDNA-templated yeast pathway, DSB repair can involve the capture of retrotransposon or reverse-transcribed RNAs in mammalian cells [98,99,100]. In addition, RNA oligonucleotides can direct the repair of a chromosomal DNA break in human cells, with a strand-bias consistent with a direct, RNA-templated mechanism [101]. Furthermore, Rad52 has been detected at DNA lesions in G0 or post-mitotic human cells, in a manner depending on transcription and DNA:RNA hybrid accumulation [96,102].

Other examples of RNA-dependent rearrangements have been reported recently. In mammals, expression of artificially-created chimeric RNAs can template translocations [103] and in plants, RNAs derived from ribozyme processing following transgenic expression, or delivered by particle bombardment, can be used as donor repair templates during CRISPR/Cpf1-mediated genome editing [104]. RNA-templated rearrangements naturally occur in ciliates, where small RNAs specify the retention or the removal of non-coding DNA sequences during the development of the macronucleus (reviewed in [5]). Since nascent RNAs are associated with the NHEJ machinery in human cells, it is possible that RNA is used as an intermediate in an NHEJ error-free repair pathway [38,105]. Whether RNAs serve as scaffolds, bridges or templates in these different situations remains to be determined.

## 7. Concluding Remarks

In this review, we pay homage to Miro Radman’s early work on DNA repair and explore how RNA-containing “mismatches” are generated, utilized or repaired by native cellular machineries to insure genome stability. As summarized, a number of enzymatic reactions naturally involved in DNA replication, repair or recombination (the 3R) appear to be the source of RNA insertions within the genome. Such RNA-containing structures have a dual potential: on the one hand, their processing by the cellular machinery can lead to genetic instability, calling for their specific removal; on the other hand, they can be used as scaffolds or templates in chromosome maintenance, with contributions to genome plasticity and regulation.

It is not surprising that RNA is a substrate for the 3R machinery, considering that DNA likely appeared later in early life forms to act as a more stable vehicle to pass genetic information from one generation to the next. In many cases, clues from the primordial interactions between DNA and RNA are still preserved in the biochemical properties of the proteins that cross-react with the hybrid molecules in today’s organisms. Perhaps the interference of highly reactive RNA moieties with DNA metabolism is intrinsically unavoidable in some circumstances, thus precluding counterselection of cross-reactivity during evolution. The solutions selected rather rely on the removal of the subsequent RNA insertions, or their utilization for dedicated transactions. As highlighted, RNA-containing substrates are still used in a number of reactions beneficial for genome homeostasis, as exemplified by the role of RNA primers in initiating DNA replication.

Although RNA-dependent repair pathways operate at low rates, requiring dedicated, artificial reporters for their detection, their existence raises the question of their physiological relevance and relationship with canonical DNA repair pathways. Evidence exists that RNA- and DNA-dependent processes compete for the repair of a DSB, as exemplified by the higher rate of RNA-templated repair in the absence of Rad51-dependent HDR [39]. It is therefore tempting to speculate that RNA-dependent repair could be of particular importance in the absence of the sister chromatid template, e.g., in differentiated or post-mitotic cells, or in pathological situations where the canonical repair pathways are not fully functional. Whether additional regulation is required to favor RNA-dependent repair in such situations remains to be investigated.

Finally, the cross-reactivity of recombination proteins with RNA-containing molecules supports the view that 3R factors may also directly interfere with RNA metabolism, e.g., transcription, processing and degradation. Conversely, altered RNA metabolism could interfere with the canonical functions of HR proteins on DNA substrates. In support of this hypothesis, accumulation of RNAs in exosome mutant cells is associated with a decrease in the generation of RPA-coated ssDNA [106]. In view of the recently reported association of RPA with RNAs [18], the reduced recruitment of RPA likely stems from its titration by the increased load of RNAs. In light of such extensive cross-talk between DNA and RNA metabolism, perhaps 3R should be re-baptized 4R: Replication, Repair, Recombination, RNA.

## Figures and Tables

**Figure 1 cells-10-01433-f001:**
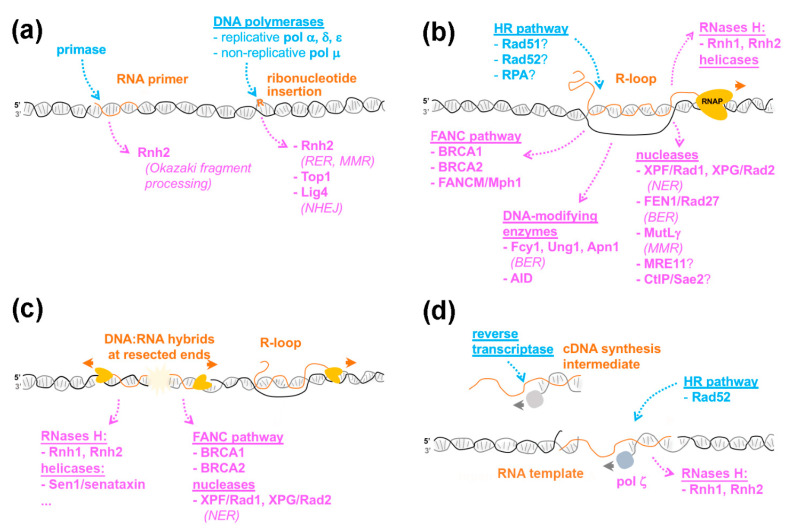
The variety of RNA-containing structures in the genome. Different types of ribonucleotide- or RNA-containing structures are represented in orange. The enzymes with reported contributions to their generation or removal are listed in blue and magenta, respectively, and the repair pathways with which they are associated are italicized. The orange and grey arrows indicate the direction of RNA and DNA synthesis, respectively. RNAP, RNA polymerase. (**a**), Ribonucleotide incorporation during DNA replication. For simplicity, only one branch of the replication fork is represented. (**b**), R-loop formation and resolution. (**c**), DNA:RNA hybrid accumulation at DSBs. (**d**), RNA-templated DNA repair. The cDNA produced upon reverse-transcription can also be used in homology-directed repair in a Rad51- and Rad52-dependent process (not depicted). See text for details.

**Figure 2 cells-10-01433-f002:**
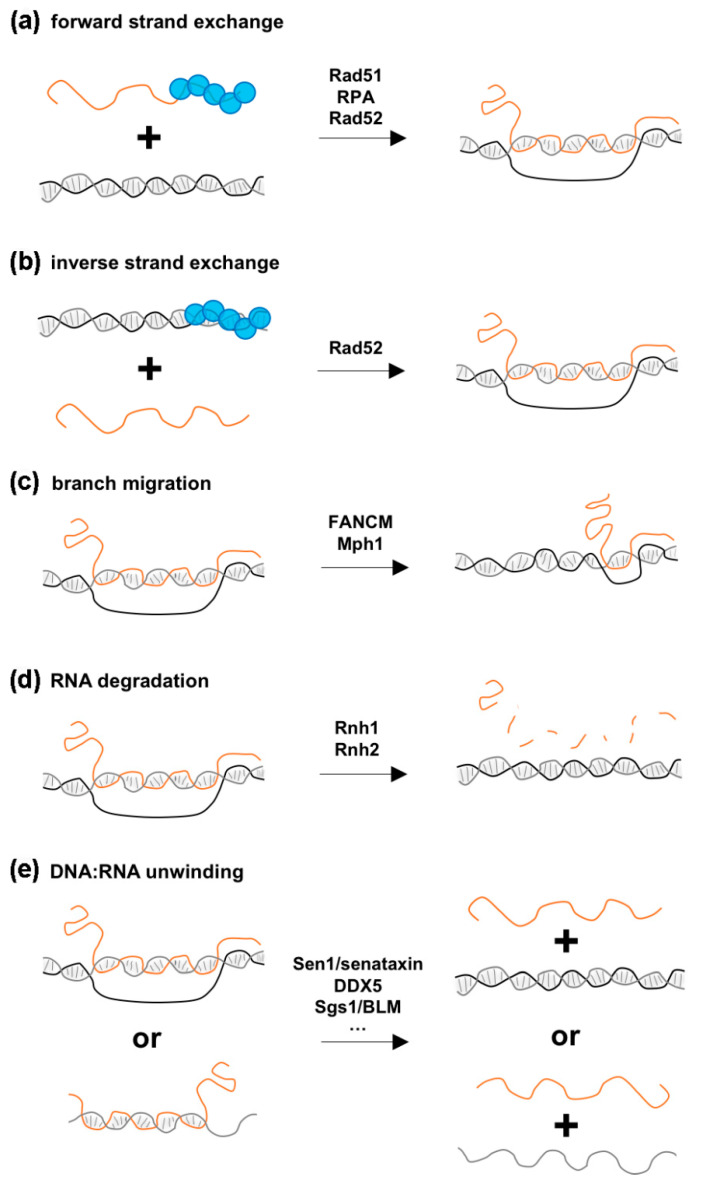
An overview of the enzymatic reactions engaging RNA-containing structures. Examples of enzymes reported to catalyze the indicated reactions are listed, and their respective substrates and products are represented for forward and inverse strand exchange (**a**,**b**), branch migration (**c**), RNA degradation (**d**) and DNA:RNA unwinding (**e**). Note that RNase H2 can also excise ribonucleotides and degrade short RNA primers embedded within the genome.

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
