# Peer review of "The Ultimate (Mis)match: When DNA Meets RNA"

_cells, 2021, doi:10.3390/cells10061433_

Round 1
Reviewer 1 Report
This is a very good and timely review. I have several optional suggestions that authors may consider at revision.
End of section 2.3:
It is brought up how ribonucleotides can affect repair and they brought up a few examples. An example not mentioned is how incorporated ribonucleotides affect HR by the selection of the resection enzyme.
Specifically, Daley, Sung 2020 ( https://www.nature.com/articles/s41467-020-16903-4 ) show how short stretches of ribonucleotides at the end of a break stimulate resection by EXO1 while resisting DNA2 cleavage. Further, long stretches of RNA and DNA:RNA hybrids inhibit resection altogether.
In section 3.3 Addressing the impact of HR-dependent R-loops on genome functions:
I would suggest adding a short statement of how R-loops at telomeres result in increased replication stress at telomeres that then induces alternative lengthening of telomeres.
Arora 2014 Nat Comm ( https://www.ncbi.nlm.nih.gov/pmc/articles/PMC4218956/ )
Lu 2019 Nat Comm ( https://www.nature.com/articles/s41467-019-10180-6 )
Silva 2019 Nat Comm ( https://www.nature.com/articles/s41467-019-10179-z )
Section 4.1 (discussing the mechanism of removal of R-loops by RNaseH2): this may not fit the scope of the review but a recent paper discusses how ADAR is involved in the RNaseH2 mediated removal of removal of R-loops at telomeres, whereupon the loss of ADAR function leads to an accumulation of R-loops.
Shiromoto 2021 Nat Comm ( https://www.nature.com/articles/s41467-021-21921-x )
Section 4.2
It would be worth note that R-loop associated C-deamination mutagenesis by AID modeled in the yeast system (ref 65) was strongly biased towards non-transcribed strand, while AID mutagenesis in IG HSM region shows practically no strand bias (reviewed in https://pubmed.ncbi.nlm.nih.gov/32434680/, see refs 72, 88, 89,90 therein).
Another potential case of ssDNA exposure to mutagenesis is cytosine deamination by ssDNA specific APOBEC3 enzymes. This mutagenesis is ubiquitous in many types of human cancers. However, deamination of non-transcribed strand of R-loop would result in fixed mutations only if normal dsDNA structure allowing templated (base) excision repair would not be restored until replication. Not surprisingly, APOBEC3 cytidine deaminases in human cancer showed no or very little transcription strand bias in whole-genome sequenced human cancers. At the same time APOBEC mutagenesis was clearly preferring lagging strand template ( https://pubmed.ncbi.nlm.nih.gov/26806129/ ). The bias towards non-transcribed strand however turned more detectable in highly transcribed genes ( doi: https://doi.org/10.1101/2021.04.14.439818 ). Additional complication to the story is recently reported correlation between transcription and replication start sites (PMID https://pubmed.ncbi.nlm.nih.gov/30598550/ ).
Section 5.3. Consider including the recent work from the Sung lab about ribonucleotides affecting resection. Daley, Sung 2020 ( https://www.nature.com/articles/s41467-020-16903-4 )
Author Response
Reviewer #1
This is a very good and timely review. I have several optional suggestions that authors may consider at revision.
We acknowledge Reviewer #1 for their interest and their useful remarks.
End of section 2.3: It is brought up how ribonucleotides can affect repair and they brought up a few examples. An example not mentioned is how incorporated ribonucleotides affect HR by the selection of the resection enzyme. Specifically, Daley, Sung 2020 ( https://www.nature.com/articles/s41467-020-16903-4 ) show how short stretches of ribonucleotides at the end of a break stimulate resection by EXO1 while resisting DNA2 cleavage. Further, long stretches of RNA and DNA:RNA hybrids inhibit resection altogether.
This work (which was previously quoted in section 5) is now also mentioned in this paragraph.
In section 3.3 Addressing the impact of HR-dependent R-loops on genome functions:
I would suggest adding a short statement of how R-loops at telomeres result in increased replication stress at telomeres that then induces alternative lengthening of telomeres.
Arora 2014 Nat Comm ( https://www.ncbi.nlm.nih.gov/pmc/articles/PMC4218956/ )
Lu 2019 Nat Comm ( https://www.nature.com/articles/s41467-019-10180-6 )
Silva 2019 Nat Comm ( https://www.nature.com/articles/s41467-019-10179-z )
Section 4.1 (discussing the mechanism of removal of R-loops by RNaseH2): this may not fit the scope of the review but a recent paper discusses how ADAR is involved in the RNaseH2 mediated removal of removal of R-loops at telomeres, whereupon the loss of ADAR function leads to an accumulation of R-loops.
Shiromoto 2021 Nat Comm ( https://www.nature.com/articles/s41467-021-21921-x )
We have chosen not to address the complex relationships between R-loop and telomere metabolism, and have restricted our discussion to the role of the HR machinery in R-loop formation at these sequences. We have nevertheless incorporated in our review the abovementioned paper by Silva et al. since it reports that FANCM can unwind telomeric R-loops (now quoted in section 4.1).
Section 4.2
It would be worth note that R-loop associated C-deamination mutagenesis by AID modeled in the yeast system (ref 65) was strongly biased towards non-transcribed strand, while AID mutagenesis in IG HSM region shows practically no strand bias (reviewed in https://pubmed.ncbi.nlm.nih.gov/32434680/, see refs 72, 88, 89,90 therein).
This is now mentioned in both section 4.2. and 4.3.
Another potential case of ssDNA exposure to mutagenesis is cytosine deamination by ssDNA specific APOBEC3 enzymes. This mutagenesis is ubiquitous in many types of human cancers. However, deamination of non-transcribed strand of R-loop would result in fixed mutations only if normal dsDNA structure allowing templated (base) excision repair would not be restored until replication. Not surprisingly, APOBEC3 cytidine deaminases in human cancer showed no or very little transcription strand bias in whole-genome sequenced human cancers. At the same time APOBEC mutagenesis was clearly preferring lagging strand template ( https://pubmed.ncbi.nlm.nih.gov/26806129/ ). The bias towards non-transcribed strand however turned more detectable in highly transcribed genes ( doi: https://doi.org/10.1101/2021.04.14.439818 ). Additional complication to the story is recently reported correlation between transcription and replication start sites (PMID https://pubmed.ncbi.nlm.nih.gov/30598550/ ).
We now mention that APOBEC enzymes could similarly target ssDNA moieties within R-loops (section 4.2).
Section 5.3. Consider including the recent work from the Sung lab about ribonucleotides affecting resection. Daley, Sung 2020 ( https://www.nature.com/articles/s41467-020-16903-4 )
This work (our reference #[32]) is now mentioned in both sections 2.3 and 5.3.
Reviewer 2 Report
This manuscript reviews recent work on the many roles of ribonucleotides incorporated into DNA. The manuscript is comprehensive, nicely covering all aspects of this circumstance. It certainly allows readers to appreciate many of the advances that have been made. I highly recommend publication in Cells, and without revision.
Author Response
We acknowledge Reviewer #2 for their positive comments.